# Agricultural land-uses consistently exacerbate infectious disease risks in Southeast Asia

Hiral A. Shah [1,2], Paul Huxley [1,2], Jocelyn Elmes [1,3] & Kris A. Murray [1,2]

Agriculture has been implicated as a potential driver of human infectious diseases. However, the generality of disease-agriculture relationships has not been systematically assessed, hindering efforts to incorporate human health considerations into land-use and development policies. Here we perform a meta-analysis with 34 eligible studies and show that people who live or work in agricultural land in Southeast Asia are on average 1.74 (CI 1.47–2.07) times as likely to be infected with a pathogen than those unexposed. Effect sizes are greatest for exposure to oil palm, rubber, and non-poultry based livestock farming and for hookworm (OR 2.42, CI 1.56–3.75), malaria (OR 2.00, CI 1.46–2.73), scrub typhus (OR 2.37, CI 1.41–3.96) and spotted fever group diseases (OR 3.91, CI 2.61–5.85). In contrast, no change in infection risk is detected for faecal-oral route diseases. Although responses vary by land-use and disease types, results suggest that agricultural land-uses exacerbate many infectious diseases in Southeast Asia.

[1] MRC Centre for Global Infectious Disease Analysis, Department of Infectious Disease Epidemiology, School of Public Health, Imperial College London, London, UK. [2] Grantham Institute—Climate Change and the Environment—Imperial College London, London, UK. [3] Department of Global Health and Development, London School of Hygiene and Tropical Medicine, London, UK. Correspondence and requests for materials should be addressed to H.A.S. (email: h.shah16@imperial.ac.uk)

Agricultural land-use and land-use change, including agricultural intensification and the conversion of forests, wetlands and grasslands into forest monocultures, crops and pasture, has led to major increases in the production of food, timber, housing and other commodities[1–3]. Although delivering economic and social benefits, these human activities have also resulted in substantial negative socio-ecological consequences, such as increased $CO_2$[4,5], air pollutant emissions[5], loss of biodiversity[6–11], modifications in surface fluxes of heat and water vapour resulting in changing regional weather patterns[12–14], degradation of air and water quality[15–17] and a decrease in the supply of renewable fresh water[18].

This trade-off between the considerable costs and benefits at stake places the agricultural sector at the heart of global sustainability, health and environmental frameworks (e.g., Sustainable Development Goals, Paris Agreement, Aichi Biodiversity Targets), and makes simultaneous achievement of key targets a formidable challenge[19].

While the impacts of agricultural land-use activities is relatively well characterised in some sectors (e.g., carbon emissions accounting frameworks[20], biodiversity loss[11,21,22]), less well established are the potential impacts on human health, where the majority of existing research signposts towards the health impacts of occupational pesticide, chemical and heavy metal exposure[23,24]. In particular, the evidence linking human-induced land-use changes and infectious disease risk outcomes in humans, many of which are related to agriculture[25–32], has not been systematically evaluated or quantified.

Numerous case studies support a link between agricultural land-use or land-use change and infectious disease risks[33]. For example, irrigation-based agriculture and rural development can expand breeding habitats of *Culex* vectors and has led to Japanese encephalitis virus establishing a secondary cycle in domestic pig populations where it amplifies and spills over into human populations[33–36]. Deforestation and associated environmental changes may facilitate the transmission of *Plasmodium knowlesi* (cause of zoonotic malaria) to humans in Malaysian Borneo[37]; expansion and changes in agricultural practices are associated with the emergence of Nipah Virus in Malaysia[38] and increased *Leptospira* infections and fatalities in Thailand have been observed in open habitats such as rice fields that are prone to flooding[39].

In addition, a number of theoretical modelling studies and meta-analyses suggest potentially generalisable links between land-use or land-use change and biodiversity loss (a key outcome of land-use change, albeit not necessarily specific to agricultural activities[31,40]), some of which may be linked to increases in disease risk. For example, Guo et al.[41] find a general increase in host or vector community competence associated with land-use changes. Rohr et al.[41] report that agricultural drivers are associated with >25% of emerging infectious diseases and >50% of emerging zoonotic infectious diseases in humans. Faust et al.[31] highlight changing host population densities and edge effects as mechanisms that could drive disease emergence in converted landscapes. Civitello et al.[43] show that host diversity inhibits parasite abundance (e.g., infection prevalence for microparasites, mean parasite load for macro-parasites, density of infected vectors for vector-borne parasites or percent diseased tissue for plant parasites) and therefore suggest that a generalisable 'dilution effect' may modulate disease risk across a number of disease systems. However, the extent to which these effects extend to human infectious diseases remain highly contentious[44], and few studies focus on specific land-use types.

Here, we test for a generalisable or net impact of occupational or residential exposure to agricultural land-use on the risk of infectious disease in humans in Southeast Asia (SE Asia) via a systematic review and meta-analysis approach, following PRISMA reporting standards for medical and epidemiological evidence syntheses.

A global review was deemed infeasible due to the vast collection of citations that would require double review to achieve PRISMA standards (~50,000 citations). We considered a narrower focus on SE Asia (defined here as the ASEAN region, including, Vietnam, Cambodia, Laos PDR, Thailand, Myanmar, Malaysia, Indonesia, Singapore, Philippines, East Timor and Brunei) as an appropriate model system given its combination of biologically diverse landscapes[8], differing land-uses[45] and because it is considered a zoonotic, parasitic and emerging disease hotspot area[46,47]. Specifically, we quantified an overall association between where people live or work in SE Asia and disease risk, finding that those in agricultural land are on average almost twice as likely to be infected with a pathogen than controls (odds ratio (OR) 1.74, confidence interval (CI) 1.47–2.07, $p < 0.001$). We also report consistent associations between forest monoculture agriculture (oil palm and rubber) and a number of specific diseases of differing ecologies and epidemiologies, while accounting for potential effects of publication bias and both within and between-study confounding. Although responses clearly vary by land-use and disease types, generalisable results from this and further studies will help identify co-management opportunities for health and the environment.

## Results

**Regional analysis.** The search strategy returned 15,426 potentially relevant publications in total, 58 of which met the inclusion criteria for full text analysis (Fig. 1). Of these, 34 mutually exclusive studies were included in the regional meta-analysis and a total of 37 mutually exclusive studies were included in the multiple subgroup analyses. Studies spanned five countries (Thailand = 11, Malaysia = 10, Vietnam = 9, Philippines = 2, Lao PDR = 2), two designs (cross-sectional = 27, case–control = 7) and were assessed as being of varying quality using two study quality tools (Office of Health Assessment and Translation (OHAT)—definitely low risk of bias = 2, probably low risk of bias = 25, probably high risk of bias = 10 and National Heart, Lung, and Blood Institute (NHLBI) —good = 7, fair = 23, poor = 4). A total of 80 effect estimates were extracted consisting of 26 infectious diseases and 12 different exposures. All included studies were in English and no studies were found to be in any other language. Full details of sample characteristics for each study including analysis groups are presented in Supplementary Data 1.

Overall, occupational or residential exposure to agricultural land-use was consistently associated with increased infectious disease risks, but effects varied widely among studies, differing disease groups and agricultural types. A regional analysis of 34 mutually exclusive crude odds ratios from 34 studies demonstrated that people exposed to agricultural land either occupationally or residentially were at a 74% increased risk of being infected with a pathogen than those unexposed (OR 1.74, CI 1.45–2.05, $p < 0.001$, $E = 2.01$, Fig. 2). Although a larger number of positive studies were included within our sample data set, as shown in the funnel plot (Fig. 3), linear regression tests and the trim and fill analyses (Fig. 2) highlighted no evidence of publication bias on the overall effect size. High between-study heterogeneity ($I^2 = 83.8\%$) was nevertheless observed, indicating considerable variability in effects among studies.

To assess the impact for within study confounding, a meta-analysis of 17 mutually exclusive adjusted odds ratios from 17 studies was conducted suggesting that people exposed to agricultural land either occupationally or residentially were similarly at significantly increased risk of being infected with a pathogen than those unexposed (OR 1.46, CI 1.11–1.92, $p < 0.001$,

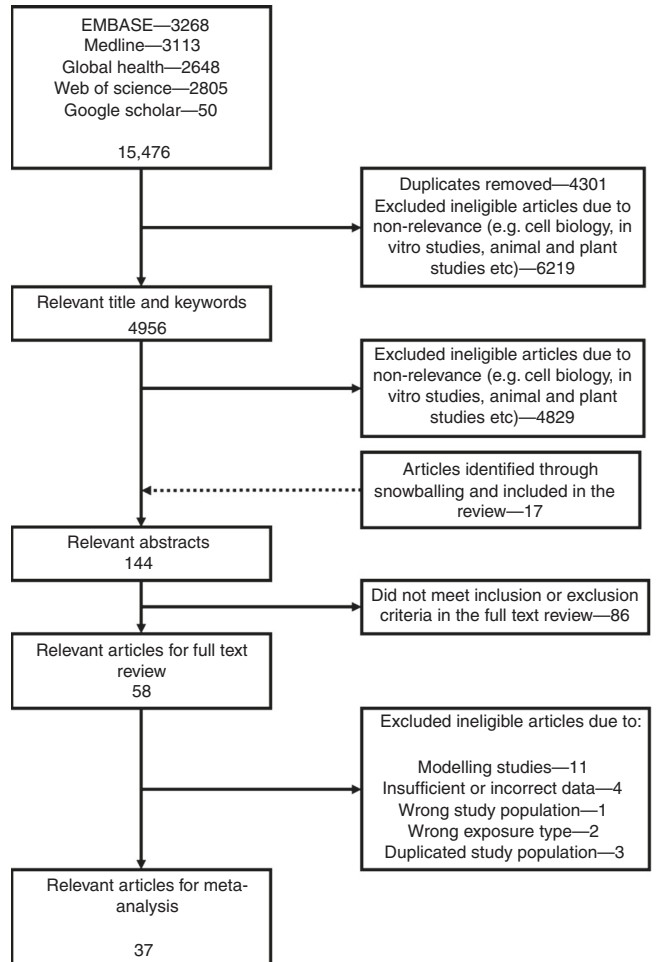

**Fig. 1** PRISMA diagram. A flow chart of the study selection process

Supplementary Fig. 1). Tests of the potential effect of unmeasured confounders suggested that an excluded variable(s) would have to have a minimum odds ratio of 2.03 with both the exposure and outcome to fully explain away the pooled result ($E = 2.03$).

**Subgroup analyses**. To evaluate the impact of between-study confounding, we examined the influence of a range of additional study and sample characteristics on effect size and direction, including study type and methodology, socio-demographic characteristics (gender, whether children were included in the sample population, and rural vs. urban), both study quality assessments and study location. In this test, associations consistent with the overall positive effect were observed irrespective of study and sample characteristics (Fig. 4), strengthening confidence that the pooled result is robust to a range of measured and, by extension, unmeasured confounders. In addition, the significant heterogeneity observed among studies in the regional pooled analysis (Fig. 2) does not suggest the presence of systematic bias from unmeasured confounders.

Nevertheless, one effect modifier/confounder variable (study setting) exhibited a divergence in effect sizes between groups, suggesting a possible interaction with the main effect of agricultural exposure. Here, the effect of agricultural exposure on infection was more than twice as strong in studies in urban than in rural settings, preserving the possibility that the pooled effect is vulnerable to the effect of unmeasured confounders, albeit here insufficient to explain away the pooled result. In addition, a single subgroup indicated a lack of significant association (studies

based in Lao PDR). However, given the effect sizes and direction for these groups did not deviate considerably from the pooled effect, we considered this more likely due to small sample size than evidence of potential confounding. Finally, low heterogeneity for some stratum specific covariates alongside consistent effect sizes indicates that the source of heterogeneity is likely coming from elsewhere, warranting the use of further subgroup analyses to scrutinise the pooled result and to test our hypotheses on differences in effect between agricultural types and disease groups.

Further subgroup analysis was performed using mutually exclusive estimates based on common exposure types (Figs. 5 and 6) and for specific disease classes (Fig. 7). Consistent associations between agricultural exposure and infection were again evident. For the non-specific agricultural group, a similar effect was observed with all infectious diseases (OR 1.71, CI 1.38–2.13). When stratifying the non-specific agricultural group by disease class, significant effects were observed for parasitic (OR 1.74, CI 1.41–2.13), vector-borne (OR 1.85, CI 1.18–2.90) and zoonotic diseases (OR 1.63, CI 1.19–2.24). A marginal non-significant effect was found for bacterial diseases (OR 1.79, CI 0.97–3.31, $I^2 = 89.4\%$) (Fig. 5 and Supplementary Table 1).

Among the specific agricultural subgroups, the effect was higher in populations working or living in or near oil palm and being infected with vector-borne and zoonotic diseases (leptospirosis and *P. knowlesi*) compared to those unexposed (OR 3.25, CI 2.29–4.61). Similarly, exposure to rubber plantations increased the risk of being infected with all types of pathogens (OR 2.27, CI 1.82–2.82). This effect was also consistent when stratified by disease class where significant associations were found for bacterial (OR 2.27, CI 1.79–2.89), parasitic (OR 2.24, CI 1.35–3.74), vector-borne (OR 2.27, CI 1.82–2.82) and zoonotic (OR 2.31, CI 1.83–2.94) disease class subgroups (Fig. 6 and Supplementary Table 1).

Significant associations were observed for general livestock farming (Fig. 6 and Supplementary Table 1) and all diseases (OR 2.54, CI 1.37–4.72), zoonotic (OR 2.46, CI 1.35–4.48), vector-borne (OR 2.52, CI 1.48–4.28) and bacterial (OR 4.47, CI 1.30–15.39) diseases. A marginal non-significant-positive association was also established between livestock farming and viral diseases (OR 1.55, CI 0.83–2.81). Further subgrouping by livestock type showed consistent marginal non-significant-positive effects. Specifically, marginal associations were observed between porcine animals and all diseases (OR 3.57, CI 0.84–15.23), vector-borne (OR 3.09, CI 0.58–16.46), zoonotic (OR 3.57, CI 0.84–15.23) and viral (OR 4.31, CI 0.49–37.81) diseases. Effect sizes found for bovine animals were consistent for all, vector-borne or zoonotic diseases (OR 2.09, CI 0.80–5.49) and bacterial diseases (OR 2.40, CI 0.57–10.12). No associations were found for exposure to poultry and all, vector-borne or zoonotic diseases (OR 0.91, CI 0.24–3.45). There was no evidence of publication bias for any other exposure-based subgroups (Supplementary Table 2).

Exposure to rice paddy farming (Fig. 5) resulted in a non-significant association for all diseases (OR 1.34, CI 0.81–2.23), bacterial (OR 1.40, CI 0.71–2.77), zoonotic or vector-borne (OR 1.17, CI 0.62–2.21) disease class subgroups. However, trim and fill tests (Supplementary Table 2) indicated the presence of publication bias in which positive associations between agricultural exposure and general infection were under-reported among studies on rice paddy farming. When accounted for, the effect of agricultural exposure on infection risk within the rice paddy farming subgroup became significant (OR 1.81, CI 1.04–3.17, $p = 0.037$ (Z-test), $E = 1.47$), suggesting that the overall effect is likely conservative.

A final subgroup analysis based on specific diseases or disease complexes again showed consistent associations between

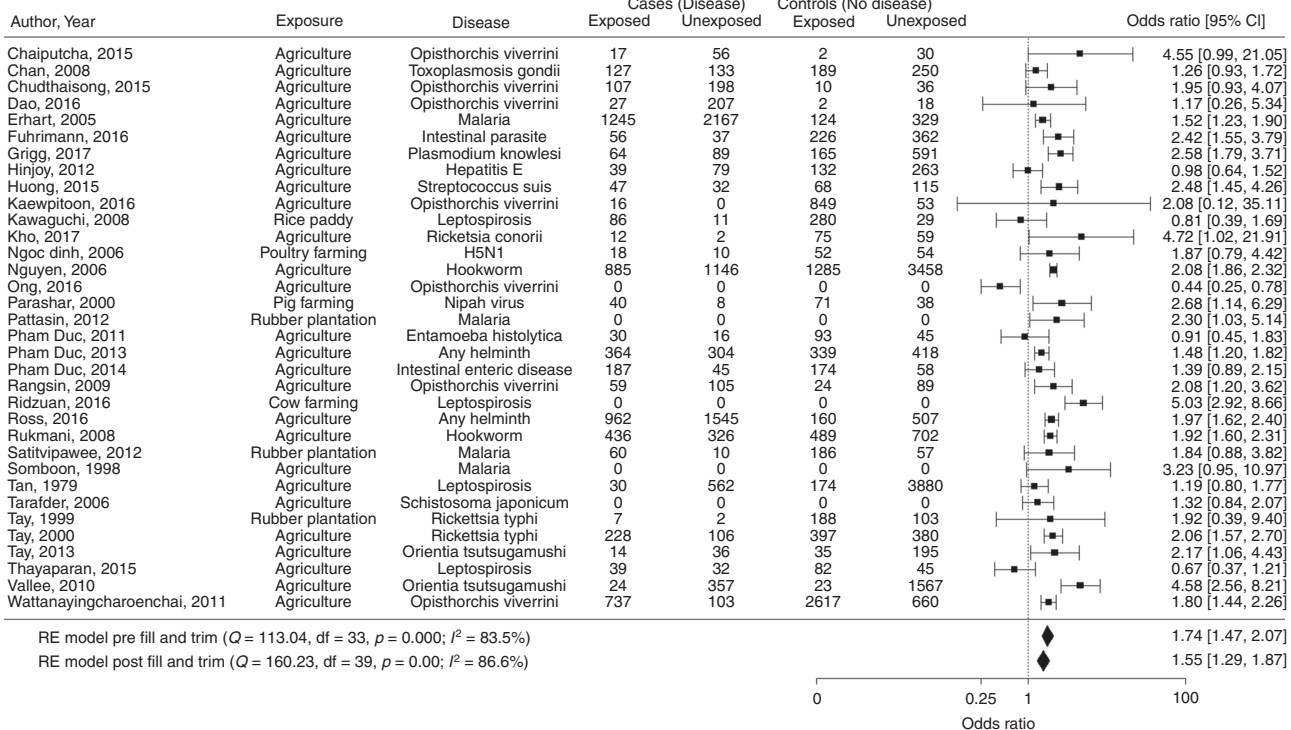

| Author, Year | Exposure | Disease | Cases (Disease) Exposed | Cases (Disease) Unexposed | Controls (No disease) Exposed | Controls (No disease) Unexposed | Odds ratio [95% CI] |
|---|---|---|---|---|---|---|---|
| Chaiputcha, 2015 | Agriculture | Opisthorchis viverrini | 17 | 56 | 2 | 30 | 4.55 [0.99, 21.05] |
| Chan, 2008 | Agriculture | Toxoplasmosis gondii | 127 | 133 | 189 | 250 | 1.26 [0.93, 1.72] |
| Chudthaisong, 2015 | Agriculture | Opisthorchis viverrini | 107 | 198 | 10 | 36 | 1.95 [0.93, 4.07] |
| Dao, 2016 | Agriculture | Opisthorchis viverrini | 27 | 207 | 2 | 18 | 1.17 [0.26, 5.34] |
| Erhart, 2005 | Agriculture | Malaria | 1245 | 2167 | 124 | 329 | 1.52 [1.23, 1.90] |
| Fuhrimann, 2016 | Agriculture | Intestinal parasite | 56 | 37 | 226 | 362 | 2.42 [1.55, 3.79] |
| Grigg, 2017 | Agriculture | Plasmodium knowlesi | 64 | 89 | 165 | 591 | 2.58 [1.79, 3.71] |
| Hinjoy, 2012 | Agriculture | Hepatitis E | 39 | 79 | 132 | 263 | 0.98 [0.64, 1.52] |
| Huong, 2015 | Agriculture | Streptococcus suis | 47 | 32 | 68 | 115 | 2.48 [1.45, 4.26] |
| Kaewpitoon, 2016 | Agriculture | Opisthorchis viverrini | 16 | 0 | 849 | 53 | 2.08 [0.12, 35.11] |
| Kawaguchi, 2008 | Rice paddy | Leptospirosis | 86 | 11 | 280 | 29 | 0.81 [0.39, 1.69] |
| Kho, 2017 | Agriculture | Ricketsia conorii | 12 | 2 | 75 | 59 | 4.72 [1.02, 21.91] |
| Ngoc dinh, 2006 | Poultry farming | H5N1 | 18 | 10 | 52 | 54 | 1.87 [0.79, 4.42] |
| Nguyen, 2006 | Agriculture | Hookworm | 885 | 1146 | 1285 | 3458 | 2.08 [1.86, 2.32] |
| Ong, 2016 | Agriculture | Opisthorchis viverrini | 0 | 0 | 0 | 0 | 0.44 [0.25, 0.78] |
| Parashar, 2000 | Pig farming | Nipah virus | 40 | 8 | 71 | 38 | 2.68 [1.14, 6.29] |
| Pattasin, 2012 | Rubber plantation | Malaria | 0 | 0 | 0 | 0 | 2.30 [1.03, 5.14] |
| Pham Duc, 2011 | Agriculture | Entamoeba histolytica | 30 | 16 | 93 | 45 | 0.91 [0.45, 1.83] |
| Pham Duc, 2013 | Agriculture | Any helminth | 364 | 304 | 339 | 418 | 1.48 [1.20, 1.82] |
| Pham Duc, 2014 | Agriculture | Intestinal enteric disease | 187 | 45 | 174 | 58 | 1.39 [0.89, 2.15] |
| Rangsin, 2009 | Agriculture | Opisthorchis viverrini | 59 | 105 | 24 | 89 | 2.08 [1.20, 3.62] |
| Ridzuan, 2016 | Cow farming | Leptospirosis | 0 | 0 | 0 | 0 | 5.03 [2.92, 8.66] |
| Ross, 2016 | Agriculture | Any helminth | 962 | 1545 | 160 | 507 | 1.97 [1.62, 2.40] |
| Rukmani, 2008 | Agriculture | Hookworm | 436 | 326 | 489 | 702 | 1.92 [1.60, 2.31] |
| Satitvipawee, 2012 | Rubber plantation | Malaria | 60 | 10 | 186 | 57 | 1.84 [0.88, 3.82] |
| Somboon, 1998 | Agriculture | Malaria | 0 | 0 | 0 | 0 | 3.23 [0.95, 10.97] |
| Tan, 1979 | Agriculture | Leptospirosis | 30 | 562 | 174 | 3880 | 1.19 [0.80, 1.77] |
| Tarafder, 2006 | Agriculture | Schistosoma japonicum | 0 | 0 | 0 | 0 | 1.32 [0.84, 2.07] |
| Tay, 1999 | Rubber plantation | Rickettsia typhi | 7 | 2 | 188 | 103 | 1.92 [0.39, 9.40] |
| Tay, 2000 | Agriculture | Rickettsia typhi | 228 | 106 | 397 | 380 | 2.06 [1.57, 2.70] |
| Tay, 2013 | Agriculture | Orientia tsutsugamushi | 14 | 36 | 35 | 195 | 2.17 [1.06, 4.43] |
| Thayaparan, 2015 | Agriculture | Leptospirosis | 39 | 32 | 82 | 45 | 0.67 [0.37, 1.21] |
| Vallee, 2010 | Agriculture | Orientia tsutsugamushi | 24 | 357 | 23 | 1567 | 4.58 [2.56, 8.21] |
| Wattanayingcharoenchai, 2011 | Agriculture | Opisthorchis viverrini | 737 | 103 | 2617 | 660 | 1.80 [1.44, 2.26] |

RE model pre fill and trim ($Q$ = 113.04, df = 33, $p$ = 0.000; $I^2$ = 83.5%) — 1.74 [1.47, 2.07]

RE model post fill and trim ($Q$ = 160.23, df = 39, $p$ = 0.00; $I^2$ = 86.6%) — 1.55 [1.29, 1.87]

**Fig. 2** Regional meta-analysis. Regional meta-analysis of mutually exclusive risk estimates to determine the association between occupational or residential exposure to agricultural land-use and infectious disease prevalence. Exposure to 'agriculture' is defined as a category where a person indicates they work or live in or near agriculture regardless of the type of agriculture. Square points show the crude odds ratio for each study, solid diamonds show the pooled meta-analysis estimates and error bars are defined as the 95% confidence interval. Note: $Q$, the Cochrane $Q$-test. Df, degrees of freedom. $p$, $p$-value. $I^2$, test for heterogeneity. RE, random effects

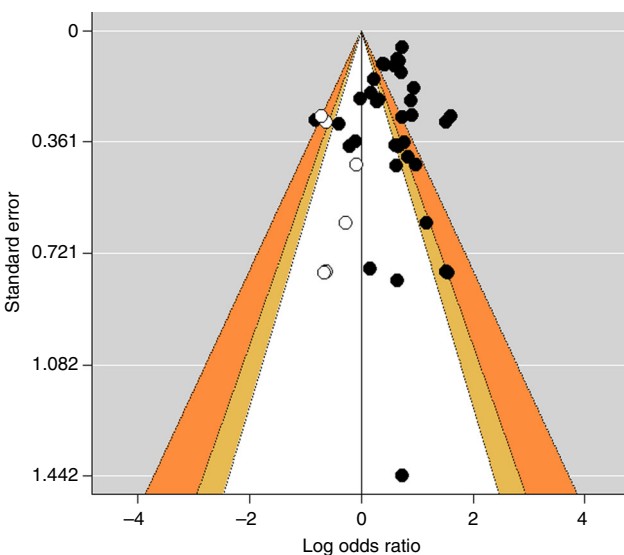

**Fig. 3** Funnel plot for the regional meta-analysis. A plot of the logarithmic risk estimates vs. the precision (standard error) for each study, with adjustment using the trim and fill method. Closed circles denote identified studies and their summary measures, respectively. Open circles represent missing studies after adjustment for funnel plot asymmetry and the summary measure incorporating hypothetical studies, respectively. Key areas of statistical significance have been superimposed on the funnel, and the plot is now centred at zero. The yellow zones show effects between $p$ = 0.10 and $p$ = 0.05, and the orange zones show effects between $p$ = 0.05 and $p$ = 0.01. Effects in the white zone are greater than $p$ = 0.10 and effects in the grey zones are smaller than $p$ = 0.01. A $Z$-test was conducted to calculate $p$-values

infection and agricultural exposure (Fig. 7 and Supplementary Table 3), notably for spotted fever group rickettsioses (OR 3.91, CI 2.61–5.85), hookworm (OR 2.42, CI 1.56–3.75), scrub typhus (OR 2.37, CI 1.41–3.96), malaria (OR 2.00, CI 1.46–2.73), *S. japonicum* (OR 1.71, CI 1.18–2.48) and *T. trichuria* (OR 1.40, CI 1.27–1.53). In contrast, no significant association was observed for the *A. lumbrocoides, O. viverrini, E. histolytica, G. intestinalis,* Leptospirosis and *R. typhi* subgroups. Again, there was little evidence of publication bias or unmeasured confounding for significant effect sizes, although heterogeneity remained present in many groups (see Supplementary Tables 1, 2 and 3 for all estimates).

## Discussion

Agricultural land-use or land-use change has been repeatedly linked to infectious disease risks in humans[25,27,28,30,31,37,41,42,48–53]; however, no study has systematically assessed or quantified this association. Based on currently available evidence from 37 eligible studies drawn from a corpus of over 15,000 peer-reviewed publications, our results strongly suggest that exposure to agricultural land-use either occupationally or residentially is consistently associated with increased infectious disease risk (average 74% increase), an effect evident across a wide range of agricultural types and disease groups. After pooling adjusted risk estimates from 17 eligible studies, a similar significant association was still evident, suggesting that there was little within study confounding.

Effects were most pronounced for oil palm monoculture (>3 times the risk) and rubber (>2 times the risk) forest monocultures and a strong association was also found for livestock farming. Associations for specific diseases or disease complexes were present for spotted fever group rickettsioses, hookworm, scrub typhus, malaria, *S. japonicum* and *T. trichuria*, but absent for

| Subgroup | n | p value | I² (%) | E value | | [95% CI] |
|---|---|---|---|---|---|---|
| **Study type** | | | | | | |
| Case control | 8 | 0.00 | 38.43 | 2.73 | | 1.93 [1.45, 2.57] |
| Cross sectional | 27 | 0.00 | 87.52 | 2.42 | | 1.71 [1.39, 2.11] |
| **Sampling strategy** | | | | | | |
| Purposive | 12 | 0.00 | 61.28 | 2.37 | | 1.68 [1.39, 2.03] |
| Random | 22 | 0.00 | 86.54 | 2.25 | | 1.60 [1.28, 2.02] |
| **Study setting** | | | | | | |
| Rural | 28 | 0.00 | 80.60 | 2.31 | | 1.65 [1.36, 1.99] |
| Urban | 3 | 0.00 | 43.71 | 4.61 | | 3.33 [1.97, 5.65] |
| National level data | 3 | 0.01 | 69.04 | 2.37 | | 1.69 [1.13, 2.52] |
| **Outcome measurement** | | | | | | |
| Laboratory diagnosis | 31 | 0.00 | 85.88 | 2.46 | | 1.74 [1.44, 2.11] |
| Clinical diagnosis | 3 | 0.01 | 0.00 | 2.27 | | 1.61 [1.15, 2.27] |
| **Study quality NHLBI** | | | | | | |
| Good | 7 | 0.05 | 93.25 | 2.70 | | 1.91 [1.01, 3.59] |
| Fair | 23 | 0.00 | 56.15 | 2.42 | | 1.72 [1.51, 1.96] |
| Poor | 4 | 0.01 | 0.00 | 1.84 | | 1.34 [1.07, 1.69] |
| **Study country** | | | | | | |
| Thailand | 11 | 0.01 | 80.40 | 2.29 | | 1.63 [1.14, 2.33] |
| Malaysia | 10 | 0.00 | 82.32 | 2.73 | | 1.93 [1.31, 2.85] |
| Vietnam | 9 | 0.00 | 57.87 | 2.42 | | 1.72 [1.44, 2.05] |
| Philippines | 2 | 0.01 | 61.14 | 2.40 | | 1.70 [1.16, 2.49] |
| Lao PDR | 2 | 0.44 | 92.37 | 2.76 | | 1.95 [0.36, 10.68] |
| **Children included in study population** | | | | | | |
| Yes | 25 | 0.00 | 71.51 | 2.28 | | 1.63 [1.39, 1.91] |
| No | 7 | 0.00 | 86.50 | 3.52 | | 2.50 [1.36, 4.62] |
| **Gender distribution** | | | | | | |
| Equal ratio | 18 | 0.00 | 73.87 | 2.07 | | 1.49 [1.22, 1.81] |
| Over 60% female | 7 | 0.01 | 85.57 | 2.69 | | 1.90 [1.19, 3.04] |
| Over 60% male | 7 | 0.00 | 73.85 | 3.43 | | 2.44 [1.69, 3.52] |
| **Study quality OHAT** | | | | | | |
| Definitely low risk of bias | 1 | 0.00 | 0.00 | 3.62 | | 2.58 [1.79, 3.71] |
| Probably low risk of bias | 25 | 0.00 | 87.65 | 2.39 | | 1.69 [1.37, 2.09] |
| Probably high risk of bias | 8 | 0.00 | 46.08 | 2.40 | | 1.70 [1.26, 2.31] |

```
0        0.25   1              100
              Odds ratio
```

**Fig. 4** Sensitivity analysis of the regional meta-analysis. A priori subgroups based on study characteristics to test the sensitivity of the regional meta-analysis. Results suggest that subgroups based on study characteristics do not significantly alter the direction of the association between occupational or residential exposure to agricultural land-use and infectious disease prevalence. Circle points show the pooled subgroup estimates and error bars are defined as the 95% confidence interval. Note: n, number of studies included in each pooled estimate. CI, confidence intervals. OHAT, Office of Health Assessment and Translation. NHLBI, National Heart, Lung and Blood Institute

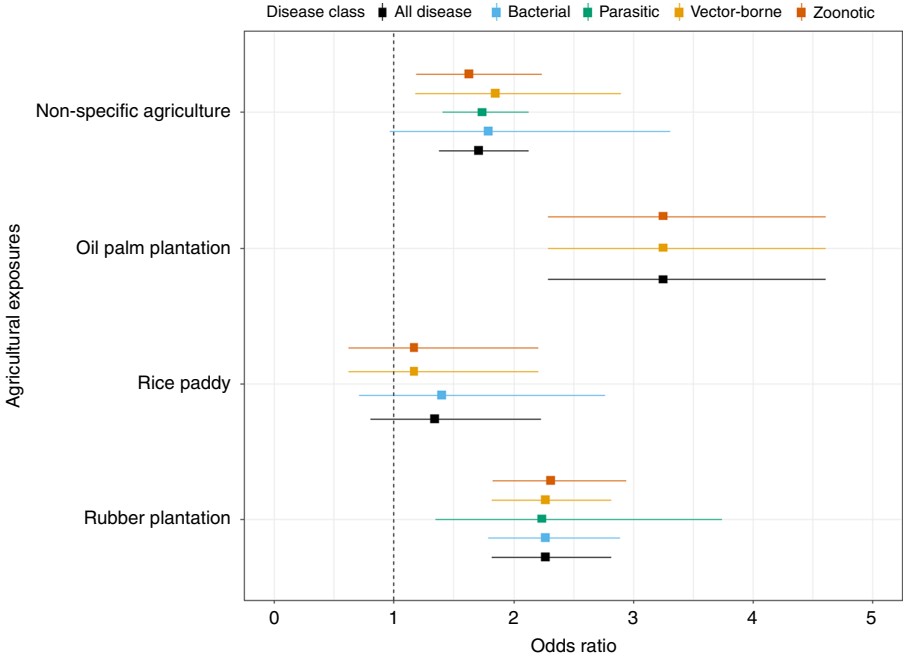

**Fig. 5** Agricultural exposure-based subgroup analysis. Subgroups were created a priori based on exposures that had two or more mutually exclusive estimates. Agriculture (non-specific) is defined as a category where a person indicates they work in agriculture regardless of the type of agriculture. Square points show the pooled subgroup estimates and error bars are defined as the 95% confidence interval

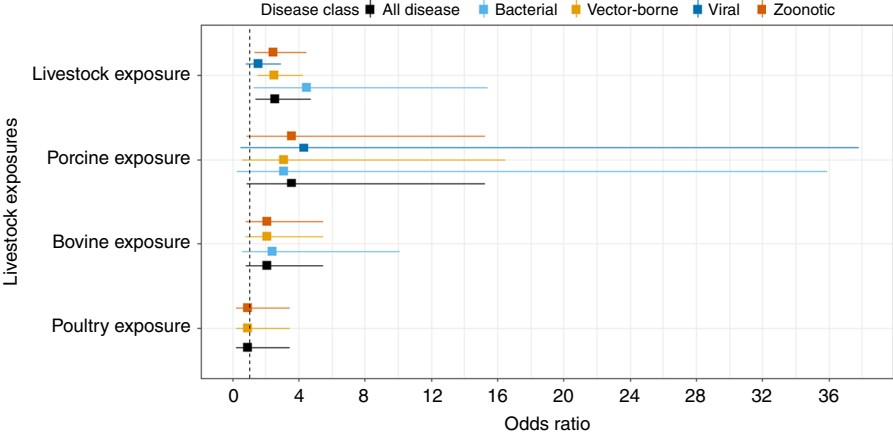

**Fig. 6** Livestock exposure-based subgroup analysis. Livestock farming is defined as a category where a person indicates they are exposed to livestock generally regardless of the specific type of livestock (e.g., chickens). Porcine, Bovine or Poultry exposure is defined as a person being exposed to each of these respective animal types. Square points show the pooled subgroup estimates and error bars are defined as the 95% confidence interval

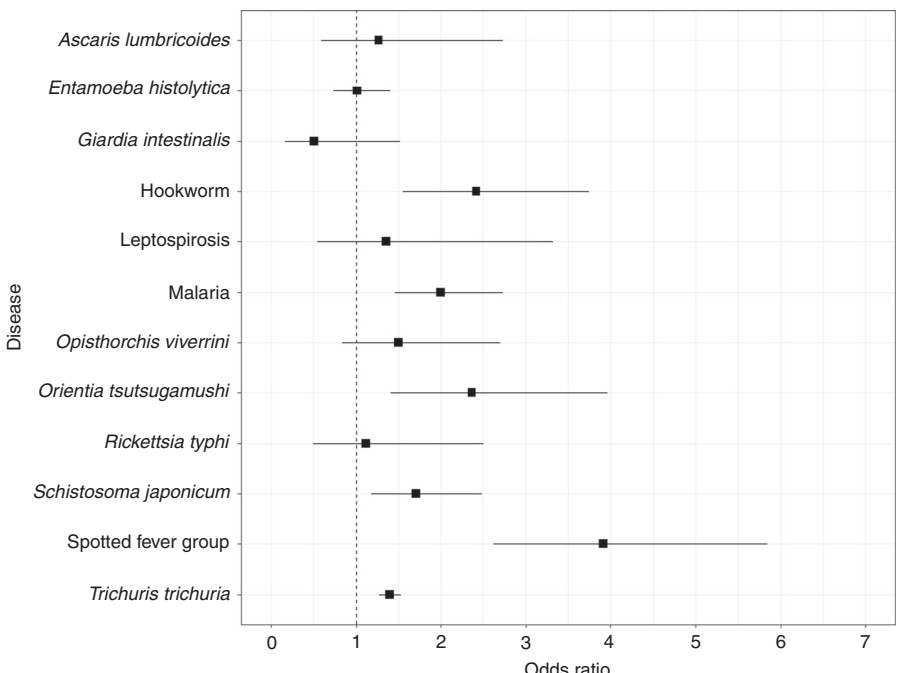

**Fig. 7** Disease-based subgroup analysis. Subgroups were created a priori based on diseases that had two or more mutually exclusive estimates. *Orientia tsutsugamushi* is also known as Scrub typhus. *Rickettsia typhi* is otherwise known as murine typhus. *Opisthorchis viverrini* is also known as Opisthorchiasis. Square points show the pooled subgroup estimates and error bars are defined as the 95% confidence interval

other groups (*A. lumbrocoides, G. intestinalis, E. histolytica*, leptospirosis, opisthorchiasis and *R. typhi*). No evidence of publication bias was detected in the regional meta-analysis, but evidence of bias was present in the rice paddy farming subgroup analysis, whereby studies documenting positive associations between agriculture and all types of infection were under-represented, suggesting the overall effect is conservative. Considerable heterogeneity among studies and subgroups alongside negative tests for potential confounding from both measured and unmeasured effect modifiers further suggest that the results are robust to a range of possible sources of bias.

Subgroup analysis, in which data were grouped by common exposures and then stratified by aetiological agent (parasitic, viral, bacterial), transmission mode (vector-borne, zoonotic) or specific disease types or disease complexes, nevertheless highlight the potential complexity and variability of agriculture-infectious disease associations. The particularly strong effects that were observed for the two-forest monoculture-based agricultural types (oil palm and rubber) are key findings. All these crops have been planted extensively in recent decades and been major contributors to land-use changes in this region. For example, between 2005 and 2010, almost 250,000 hectares of natural vegetation with tree cover was converted to rubber plantations in SE Asia[54,55], and the loss of primary forests for the cultivation of oil palm in Indonesia (especially on Sumatra and Borneo islands) quadrupled between 2000 and 2012 to 800,000 hectares a year[56]. In 2010, with an estimated 122 million people working in agriculture in SE Asia, ~115 million hectares (approx. 28% of the total area) were

harvested for rice, maize, oil palm, natural rubber and coconut[57,58]. Our results thus have far reaching implications for a large fraction of SE Asia currently under cultivation or planned for agricultural conversion; that agricultural land-uses and even differing agricultural types appear to exacerbate infectious disease risks more than others raises the possibility that land-use decisions could be tailored to minimise human health impacts.

Mechanisms by which crop monocultures impact the risk of infectious diseases are difficult to untangle and likely idiosyncratic. Deforestation or different agricultural land-uses may favour some disease hosts or vectors (influencing e.g., abundance, distributions or transmission dynamics), while the loss of biodiversity has also been linked to increases in disease risk in some cases[48]. For example, a decrease in wild mammal species richness in fragmented habitats was associated with a higher seroprevalence of Chagas disease in small mammal reservoir hosts[27,59]. In other cases, different agricultural land-use types could be frequented by people, modifying contact rates with animal hosts or vectors, and combinations of these effects are also probable. Fornace et al.[37], for example, show that a higher incidence of P. knowlesi is associated with larger amounts of forest loss surrounding villages, which may have caused changes in macaque or mosquito habitats in addition to increased levels of human activity, thereby increasing the risk of infection in humans.

Landscape factors such as distribution, density, behaviour and population dynamics of vectors and their hosts are partially controlled by landscape features such as vegetation cover, surface moisture, topography or soil type, which in turn may also influence the level of transmission of an infection[28]. Oil palm, rubber plantation and rice paddy monocultures have reduced species richness compared with primary and secondary forests[60,61], and these monocultures are structurally less complex than natural forests typically exhibiting a more uniform age structure, lower or no canopy, sparse undergrowth, less stable and more extreme microclimates, and greater levels of human disturbance and presence[61,62]. Evidence suggests that such changes related to physical characteristics of the landscape or biodiversity loss itself could favour disease carrying hosts or vectors or increase the efficacy of disease transmission to remaining hosts (in this case people). For example, Burkett-Cadena and Vittor[63] suggest that an increased mosquito vector abundance was positively associated with deforestation. Of the mosquito species that were favoured by deforestation, 56.5% were confirmed vectors of human pathogens, compared to 27.5% of species that were negatively impacted by deforestation. Faust et al.[31] also suggest that the greatest risk of spillover events occur at intermediate levels of habitat loss, whereas the largest, but rarest, epidemics occur at extremes of land conversion. Our results are thus consistent with these previous empirical[64,65] and modelling studies[31,37,49–51], and further support suggestions that deforestation resulting in crop monocultures is particularly problematic for elevating infection risks in susceptible nearby populations.

Whereas many previous studies have focussed on land-use change and deforestation explicitly, our analysis is largely blind to prior land-cover history. We nevertheless find variation in disease risk among specific agricultural land-use types, suggesting that an effect on disease risk likely goes beyond simply a change in land cover (e.g., from forest to crop monoculture) to include the final characteristics of modified agricultural landscapes. To further untangle mechanisms here would require a more detailed data set on land-cover history (e.g., class transitions), scale and context.

Previous research on the association between livestock farming and infectious disease risk has been inconsistent[39,66–72], whereas here we find consistent associations between infectious disease risks and exposure to livestock farming. Our subgroup analysis for separate livestock categories (Fig. 5) suggests that infection risk may vary according to exposure to the type of animals farmed, with pigs and cattle exposure being positively associated with infection while poultry exposure having no association. Our results further show consistent positive associations between livestock farming and differing disease classes, whereby exposure to livestock can result in two to four times the risk of being infected with vector-borne, bacterial or zoonotic diseases. We also find a marginal association with livestock farming and viral diseases, albeit with small sample sizes likely limiting power to confirm the positive effect. Livestock disease transmission can occur through multiple routes, including airborne, direct faecal-oral, animal bites and scratches, contaminated animal products and consumption of uncooked meat[70,73]. Alternatively, the impact of livestock may be to act as amplifier hosts[74,75], while livestock housing studies show that keeping livestock, such as cattle in the house as opposed to shelters outside the house contributes to increased disease risk rather than zooprophylaxis[76,77]. In addition, global changes in climate, agricultural intensification and expansion for livestock, trade, travel and closer interactions with livestock have facilitated infectious disease transmission[78]. Further empirical data from appropriately powered epidemiological studies are required to confirm our results and better identify mechanisms.

Effect variability was also observed among specific disease or disease complex subgroups. Significant associations ranging between 1.4 and 2.9 times the risk of infection when exposed to agricultural land-use were identified for hookworm, malaria, scrub typhus, S. japonicum, spotted fever group rickettsioses and Trichuris trichiura. In contrast, no effect was seen for A. lumbrocoides, E. histolytica, G. intestinalis, O. viverrini, Leptospirosis and R. typhi. These results again illustrate the potential complexity of agriculture-disease associations, whereby agricultural land-use could be impacting the transmission cycles of these disease groups in different ways or otherwise unmeasured effect modifiers could be at play.

Specific disease traits or epidemiological characteristics likely explain these differences, at least in part. For example, previous research suggests that arthropod vectors, such as mosquitoes and ticks, and helminths may be more vulnerable to environmental changes, such as agricultural land-uses than other taxa[67]. Since we find significant associations only for parasitic or vector-borne diseases (and no association for directly transmitted zoonotic or faecal-oral route diseases) our results broadly support this suggestion. Mechanistically, this may be linked to the modification of environmental niches, changes in the community composition, or alterations in the behaviour or movement of vector species[26,27,52,62,79]. For example, malaria in the Mekong region has been associated with dense forest cover and also with cultivated areas[80,81]. Forest-fringe and deforested regions can also create suitable habitats for malaria vectors (e.g., Anopheles minimus)[82]. Therefore, the wide mosquito vector diversity and the potential for mosquito vectors to adapt in deep-forests and forest-fringes, in addition to the movement of susceptible humans to and from the forest, provide ideal conditions for sustained and novel transmission[81].

Despite this trend, some diseases for which no effect was observed were helminths, and in this case variation in effect may be related to subtler transmission characteristics or other unmeasured confounders. A. lumbrocoides or O. viverrini, for example, are transmitted via the faecal-oral route, whereas T. trichuria, S. japonicum and hookworm are transmitted through skin penetration. Although both cases and controls will be infected via the same transmission mechanism, people exposed to agriculture may be more susceptible to infection with faecal-oral route transmitted diseases due to the use of night soil (human

faeces) as fertiliser to improve crop yield. Using night soil as fertiliser is prevalent in SE Asia, although there are no estimates on how widespread it may be[83–85], making it difficult to include explicitly as a potential confounding factor. Similarly, variation in effects between diseases could be a result of differential responses to public health interventions. For example, the efficacy of praziquantel mass drug administration is higher for *A. lumbrocoides* compared to *T. trichuria* or hookworm[86], but again incorporating treatment history as a potential effect modifier was not possible here.

Results show significant associations between exposure to agriculture and spotted fever group rickettsioses or scrub typhus, but not *R. typhi*. This difference could again be linked to transmission characteristics. Although all are vector-borne, both spotted fever rickettsioses and scrub typhus are tick-borne typhus-based diseases, while *R. typhi* is flea borne. This is in line with current research that suggests ticks are highly susceptible to environmental change[87,88]. For example, Lyme disease (a tick-borne disease) has increased with forest fragmentation in North America[89–91]. Ostfeld et al.[92] also find that tick-borne infection prevalence was lowest when forest cover within a 1 km radius was high. We find very little research to suggest environmental change as having large impacts on flea borne diseases[87,88]. Previous research does suggest that *R. typhi* is largely an urban disease where overcrowding, poor public health and sanitation measures are considered key risk factors for transmission[93]. Specifically, *R. typhi* typically thrives in markets, grain stores, breweries and garbage depots where rats serve as the main reservoir, which may explain the lack of association with agriculture reported here[93].

Our results also contrast with previous studies in the case of agriculture and leptospirosis. Whereas we found no overall effect for leptospirosis, previous studies have yielded mixed results[94–96]. Research conducted in Thailand suggests that the sources of human and rodent infections are different, where humans are infected in villages in non-forested areas located near rivers while rats are infected in forest patches situated in the hilly areas[39]. In Asia, humans are known to be infected through prolonged contact with water that may be contaminated by infected animal hosts[97,98]. Such environmental transmission is directly linked to frequent occupational exposure to agricultural land-use and establishing causal pathways between the environment, animal hosts and human risk is therefore required for such complex eco-epidemiologies.

Although we find a consistent association between agricultural land-use and infectious disease risk in humans, there are several inherent challenges in resolving agriculture-disease associations and some limitations in this study that could be improved upon or resolved in future studies.

First, despite the diverse range of generally robust results reported in this study, our systematic assessment of study quality does highlight an apparent lack of robust and high-quality studies that assess the impact of differing agriculture types, the degree of exposure to agriculture (e.g., more or less) and land-use change on infectious disease risks in SE Asia. Considering an initial 15,476 articles were generated from a sensitive and specific search strategy, just 34 (0.2%) met the inclusion and exclusion criteria and were included in the regional meta-analysis. All retained articles focus on agriculture as the main land-use types, as opposed to other conventional land-use practices, such as road building, dam building, mining and urbanisation. Only a small number of studies focus on the final human health outcome, while in contrast many studies focus on infectious diseases in plants or animals[52,99–101]. Similar research aiming to evaluate the impacts of agricultural land-use on biodiversity appears far more prevalent and incorporates a wider range of land-use

types[7,22,102,103]. Caution is therefore advised in interpreting our results so as to avoid generalisations not supported by the data.

In addition, studies in the meta-analysis were all either case control or cross-sectional studies, which, in the hierarchy of evidence within the medical sciences, are considered more prone to bias and confounding than some other study designs (i.e., cohort studies or randomised controlled trials)[104]. Nevertheless, most of the studies were evaluated to have probably low risk of bias or be of fair quality, indicating that there is only a small chance that a fatal flaw would invalidate an individual study's findings. Despite this, we identify a general paucity of the highest quality studies on the human health implications of land-use decision making and policy, and its impacts on infectious diseases. Further studies that capture bias, confounding and effect modification would be particularly valuable.

Second, we were not able to determine whether the associations are significant spatially and temporally or if the associations are transient. Understanding whether the association between land-use and infectious disease is consistent both spatially and temporally is an important avenue for future research. Specifically, understanding the causal relationships, leading from distal environmental changes to alterations in more proximal environmental characteristics and disease transmission cycles, which eventually lead to a shift in the risk of infectious diseases at the landscape level[53] should be prioritised for future research.

Third, although we made extensive efforts to control (through our inclusion/exclusion criteria and the subgroup analysis) or at least detect (through tests of heterogeneity, the meta-analysis of adjusted odds ratios and *E*-score tests) the potential effect of confounders and effect modifiers, there are likely to be environmental, social, demographic or even economic factors that could impact the association between land-use and infectious disease risks. Participatory epidemiology offers the opportunity to conduct bottom up agro-system analytical research on the patterns of diseases in animal and human populations[105–107]. Participatory epidemiological research has previously provided insights into how social factors (which can be potential confounders or effect modifiers) can impact ecological processes. For example, the involvement of women in the care and preparation of poultry carcasses in Egypt could contribute to higher incidence of highly pathogenic avian influenza in women[106,108]. Similarly, understanding how local indigenous herder knowledge on the clinical signs of classical acute and milder rinderpest has previously aided in the control and eradication of rinderpest[106,109]. Hence, participatory mixed methods research is an ideal platform to assess effect modification and confounding and their potential impact on disease-agriculture relationships.

Finally, substantial heterogeneity was also observed in our regional meta-analyses, where $I^2$ values were >80%. The substantial heterogeneity may be due to clinical heterogeneity or statistical heterogeneity. Clinical heterogeneity occurs where the exposure is modified by factors that vary across studies, the type of exposure (e.g., different agricultural types—rice vs. rubber) or study participant characteristics[110]. Differences between studies in the definition or the measurement of exposure or outcome, may all lead to a difference in effects. In contrast, statistical heterogeneity exists when the true effects being evaluated differ between studies and may be detectable if the variation between the results of the studies is above that expected by chance[111]. Further subgroup and sensitivity analysis showed that heterogeneity decreased to a moderate level ($I^2 < 60\%$) only for certain subgroups[111,112]. This suggests that some of the observed heterogeneity is attributable to epidemiological and environmental differences within this subgroup[111,113]. There was little evidence of significant publication bias in our analyses (except for rice

paddy farming), and any publication bias that was present had very little impact on the pooled association.

This meta-analysis provides broad evidence that occupational or residential exposure to differing types of agriculture can consistently exacerbate infectious disease risks in humans in SE Asia. These trends suggest that further expansion or intensification of land-use for agricultural purposes may result in the novel emergence of pathogens as observed elsewhere (e.g., refs. [30,38,50,114,115]) or increased transmission of zoonotic, parasitic or vector-borne diseases (e.g., refs. [37,51,81]). However, the results presented in this study also provide an opportunity for land-use decision makers, governments, companies and agriculturalists to recognise the impact that agricultural land-use or land-use change may have on susceptible populations and proactively identify measures to mitigate these risks.

Given a range of other negative externalities of agriculture identified in other fields (e.g., carbon emissions, air pollution, biodiversity loss), the potential for better land-use decisions to collectively minimise infectious disease impacts alongside these other impacts is large. Enhancing the sustainability of agriculture has already been identified as a nexus issue that is central to meeting a diverse range of development and environmental targets, such as the SDGs, the Aichi biodiversity targets, and the Paris agreement[19]. Key measures are already being proposed to sustainably meet this multiplicity of demands through policy changes, such as reducing food wastage throughout the food supply chain[116,117], advocation of reduced emissions and more sustainable diets[118,119], efforts in soil management techniques[120], responsible consumption of animal products[120] and biodiversity-friendly farming practices[121]. Our study provides critical additional evidence to propel human health impacts from infectious diseases into this mix to further advance health targets (e.g., SDG3, Target 3.3)[122] as a central component of improving the sustainability of agricultural development more broadly.

## Methods

**Search strategy and selection process**. Following PRISMA protocol and reporting standards for systematic reviews, we independently and systematically screened articles in April 2017 using five academic literature databases: Medline, PubMed, Global Health, Web of Science and EMBASE alongside Google Scholar.

Search strings were created through a PECOS statement using three categories (exposure, location and outcome) with Boolean operators AND between categories and OR within categories. Where applicable, MeSH terms for communicable disease, SE Asia, land-use and agriculture were also used. Differing land-use types were incorporated into the search strategy to improve the sensitivity of the search. To improve the specificity of the search strategy, the location category was only applied for title and abstracts, to capture all publications that had a study context within SE Asia. No language restrictions were placed within the search strategy. An example of the search strategy can be found in the Supplementary Note 1.

Articles were initially assessed for relevance first by title, as well as keywords if these were available, then by abstract and finally by full text. We simultaneously assessed the suitability of the studies retained after screening for full text analysis for their potential inclusion in meta-analyses, rejecting studies for which risk or odds estimates could not be calculated. Disagreements were resolved by consensus, and where no consensus was achieved a third investigator was consulted. One reviewer (H.S.) then extracted outcome and exposure data as well as data on population and study characteristics into a bespoke data extraction framework, which was then validated by a second reviewer (P.H.)[112].

**Eligibility**. Following PRISMA guidelines and the PICOS framework, we considered the following factors to determine eligibility criteria: 'study question', 'populations', 'exposure', 'comparators' and 'outcome'. A description of each follows.

Study Question—Is there an association between occupational or residential exposure to agricultural land-uses and being infected with a pathogen for adults aged 18 and above in SE Asia?

Study Design—Empirical observational studies (longitudinal cohorts, case control or cross-sectional) studies conducted in the Association of Southeast Asian Nations (ASEAN) region and reported in English were considered eligible. We anticipated that the extent and effects of language bias may have diminished recently because of the shift towards publication of studies in English[123]; however,

we reserved the option to have non-English articles translated to bolster sample sizes if a reasonable number of non-English studies were found.

Populations—This study drew participants from the general adult population aged 18 and above in SE Asia. Studies that recruited participants of all ages (including children) were also included. Studies that focused exclusively on the child population were excluded.

Exposure—The primary exposure of interest was defined as occupational or residential exposure to agriculture or agricultural land-use. This was defined as whether study participants would be working or living in or near agricultural land. Specifically, agricultural exposure was defined as any person who partakes in the cultivation of land and breeding of animals and plants to provide food, fibre, medicinal plants and other products either for domestic, residential, occupational or economic purposes[88].

Comparators—Studies were included if they compared outcomes in the exposed group with those in a group of unexposed people (people who are not occupationally or residentially exposed to agriculture or agricultural land-use).

Outcome—Studies were included if one of the primary outcomes include prevalence, seroprevalence or incidence for all infectious diseases that have a biologically plausible link to agriculture or agricultural land-use.

Studies that investigated non-communicable disease or infectious diseases of plants, invertebrates or fish were excluded. We also excluded studies that were not based on SE Asia, did not include some form of land-use as an exposure or study focus, were theoretical research papers, reviews, commentaries or letters, or were not published in English (following determining that few non-English studies meeting all other criteria were available, see above). Studies that presented odds ratios based on the co-infection of >1 disease were excluded as co-infection could increase susceptibility to other infectious diseases[124]. Studies that assessed the impact of using human faeces (night soil) as fertiliser in agriculture were also excluded[83–85]. This is because using human faeces as fertiliser was not considered a land-use but rather a confounding behavioural activity. Studies that assessed risk factors of disease in children were also excluded[125,126] as children may be exposed to agricultural work but may also be more susceptible to certain diseases. An explicit bulleted inclusion and exclusion criteria can be found in the Supplementary Note 2.

**Study quality**. A methodological study quality assessment was conducted using two quality appraisal tools sourced from the OHAT and the NHLBI Quality Assessment website.

The first tool was the OHAT Risk of Bias Rating Tool for Human and Animal Studies, which evaluates the assessment of whether the design and conduct of the study compromised the credibility of the link between exposure and outcome. The OHAT for human studies contains 11 risk-of-bias questions that cover six different domains, including selection, confounding, performance, attrition/exclusion, detection, and selective reporting bias. Six of the 11 questions are applicable for cross-sectional and case control studies and are answered using one of four predefined answer choices (1) definitely low risk of bias; (2) probably low risk of bias; (3) probably high risk of bias; and (4) definitely high risk of bias. Studies were excluded from this review if they had an average rating of definitely high risk of bias and/or if there was substantial evidence that the studies showed threats to internal validity.

The second set of tools were for Observational Cohort and Cross-Sectional Studies (QAT—OCCSS), and for case control studies (QAT—CCS). Both tools had 14 and 9 items, respectively, that classified study quality using specific epidemiological parameters, such as transparency of research question, sources of potential bias (e.g., selection or measurement), study power, confounding and other items that inferred internal validity of each study[127,128]. A greater number of Yes responses indicated a higher study quality for both study quality tools. Studies were classed as good if they presented information on all key criteria within the tools such as: research question, study population, sample size justification, exposure measurement and outcome measurement. Studies were classed as fair if they presented some information on the key criteria. Poor studies were classed as studies that could not satisfy the majority of key criteria.

**Data synthesis and statistical analysis**. Data were summarised as the number of individuals with and without infection stratified by whether they were exposed to agricultural land-use or not. Associations were quantified using the odds ratio (OR) with a 95% confidence interval. This was extracted where possible from the studies or self-calculated using relevant data where possible. Where ORs could not be extracted or calculated due to poor or non-reported data, studies were excluded from the meta-analysis[112].

A regional meta-analysis was conducted with a random effects model[110,129] to calculate a pooled estimate that quantifies the overall impact of how any occupational or residential exposure to agricultural land-use impacts the odds of infectious disease prevalence. For this, we selected mutually exclusive studies and odds estimates to be incorporated into the regional meta-analysis. This was to avoid any double counting of estimates, which could otherwise bias pooled estimates. Only one estimate was used per study and other estimates from the same study population were excluded. This was achieved by systematically selecting risk/odds estimates based on agriculture as a general occupational exposure. However, in some cases, studies provided multiple agricultural exposures or multiple disease

outcomes in the same study (e.g., oil palm, rice, rubber as an exposure type or hookworm, *T. trichuria* and *A. lumbrocoides* infection as the outcome). In these types of studies, we selected the exposure and outcome that had the largest number of cases to maximise study power. In some instances, there were multiple publications by the same author analysing the same study population[130–132]. In these cases, only the most recent publication was selected for incorporation into the overall analysis.

Random effects meta-analyses assume that a distribution of effects exists across all studies included in the analyses, resulting in heterogeneity among study results. The use of a random effects model was considered appropriate here because we assume that the associations between occupational or residential exposure to agricultural land-use or land-use change and infectious disease risks are likely to be inconsistent and idiosyncratic, which might otherwise bias the results. Therefore, we considered a random effects meta-analysis to be a more conservative approach than fixed effects analysis[110,129].

All analyses were conducted in R version 3.2.5[133] with the metafor package[134].

**Heterogeneity and subgroup analysis**. We first tested heterogeneity of effect sizes among studies included in our overall analysis using the $I^2$ statistic and the Cochranes Q-test. A value of >75% for the $I^2$ statistic is generally considered to suggest substantial heterogeneity[111,113].

We performed a subgroup analysis to determine how robust the regional meta-analysis result would be to certain study characteristics using the estimates from the regional meta-analysis. Here we created a priori subgroups on study type, sampling strategy, study setting, outcome measurement, study quality, study country and the characteristics of the study population.

Subgroup analyses were conducted on common exposures stratified by aetiological agent (parasitic, viral, bacterial) and transmission mode (vector-borne, zoonotic) or specific disease or disease complex subgroups that had more than two mutually exclusive estimates available. In order to preserve sample sizes and remain epidemiologically realistic, aetiological agent and transmission mode subgroups were not constrained to be mutually exclusive (e.g., a disease can be both vector-borne and zoonotic, such as zoonotic malaria).

Common exposures that had more than two estimates included non-specific agriculture (defined as a category where a person indicates they work in agriculture regardless of the type of agriculture), livestock farming, oil palm plantation work, rice paddy farming and rubber plantation work. Livestock farming was further stratified into common livestock groups, including porcine, bovine and poultry related exposure. Common diseases that had more than two risk estimates included *Ascaris lumbrocoides*, *Entamoeba histolytica*, *Giardia intestinalis*, hookworm, leptospirosis, malaria, *Opisthorchis viverrini*, scrub typhus (*Orientia tsutsugamushi*), *Rickettsia typhi*, *Schistosoma japonicum*, spotted fever group and *Trichuris trichiura*.

**Confounding**. We were unable to adjust our pooled regional meta-analysis estimate for known confounders and effect modifiers due to lack of individual participant level data. However, we conducted a meta-analysis of adjusted odds ratios extracted from each study to assess the potential impact of within study confounding.

In addition, considering that the association between land-use and infectious disease may be impacted by many variables that are unmeasured or unreported in published articles (e.g., temperature, rainfall, climate, soil type, topography, socio-economic status), we conducted a sensitivity analysis using an E-value to test for between-study unmeasured confounding. The E-value represents the strength of association an unmeasured confounder would need to have with both the treatment and outcome to fully explain away a specific risk factor-outcome association[135]. The E-value is calculated using the following equation: E-value = $OR + \sqrt{OR \times (OR - 1)}$[135].

When calculating the E-value, unmeasured confounders are not listed and tested explicitly. Additionally, the E-value, does not assess measurement or selection bias. The E-value results also do not guarantee that if a confounder with parameters of a particular strength exists, then it necessarily explains away the effect. Rather, it is, only possible to construct scenarios in which it could. Readers and other researchers may then assess whether any confounding associations of that magnitude are biologically plausible[135].

**Publication bias**. We assessed publication bias in three ways. First, we plotted individual study effect sizes against the standard error of each study as a measure of the study size in funnel plots to visually assess asymmetry[136]. Second, we tested this asymmetry using Egger's linear regression test, in which significant asymmetry would suggest bias or heterogeneity[137]. Finally, we used a trim and fill method to further assess if there was a likelihood of missing studies that might exist and whether this would impact the pooled estimate. This method imputes hypothetical negative unpublished studies to mirror the positive studies, and recalculates a pooled estimate to assess the impact these hypothetical studies have on the pooled effect size[138,139].

**Reporting summary**. Further information on research design is available in the Nature Research Reporting Summary linked to this article.

## Data availability

The authors declare that all published data collated during the systematic review supporting the findings of this study are available within the paper and its Supplementary Information files. The final data set is presented in Supplementary Data 1 and Supplementary Note 3. This data set presents information extracted by the reviewers and highlights the estimates used for each analysis. A description of the data set is presented in Description of Additional Supplementary Files.

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

## Acknowledgements

We thank Julie Wendling for research support. We thank Julia Dunn (Imperial College London) for her insights into the epidemiology and control of helminths. We also thank the many experts that provided constructive feedback on earlier drafts of the manuscript, including Professor Paolo Vineis, Dr Roman Carrasco, and members of the EXPO-sOMICS and Malaria Modelling teams at Imperial College London. We also acknowledge joint Centre funding from the UK Medical Research Council and Department for International Development (MR/R015600/1). H.S. is a Grantham Institute and Commonwealth Scientific and Industrial Research Organisation (CSIRO) funded Ph.D student with the Science and Solutions for a Changing Planet Doctoral Training Partnership at the Grantham Institute, Imperial College London. P.H. is a Natural Environment Research Council (NERC) funded Ph.D student with the Science and Solutions for a Changing Planet Doctoral Training Partnership at the Grantham Institute, Imperial College London.

## Author contributions

H.S., J.E. and K.M. designed the study. H.S. and P.H. conducted the systematic review. H.S. wrote the modelling code, conducted the analysis and generated the figures. H.S. and K.M. wrote the manuscript, and all authors contributed to edits and revisions.

## Additional information

**Competing interests:** The authors declare no competing interests.

