## [Peer Review File · Nature Communications]

Reviewers' Comments:

Reviewer #1:

Remarks to the Author:

This is an important study examining land-use change in Southeast Asia and impacts on disease risks in human populations. The results are very interesting: overall, the authors find a 65% increase in disease risk among agricultural areas; however, this is substantially elevated in oil palm and rubber plantations, two of the dominant agricultural crops in the region. These findings are informative scientifically, and could also be practical if policymakers heed these findings to change development strategies in order to minimize disease risk in this critical region, Southeast Asia.

My main concern is the small sample size: with only 34 studies included in the analysis. But I don't think that necessarily prevents this study's publication.

I would've liked to see more about the relationship between the different variables (e.g., crop type vs. disease type), see Gibson et al. Nature Figure 3 (which should be cited, see below) for a good example. Can you add something examining the relationship between different variables?

What is going on with the Methods section? You have 3 parts – the first "Search Strategy and selection process" using normal sentences, the second "Eligibility" as a list, the third "Inclusion Criteria" as bullet points – all with the same information. Only one is needed.

Minor comments:

2, 49, 118, 434, 444, 532: "South-East Asia" or "South East Asia" or "south-east Asia"? Be consistent. The most commonly used is something else: "Southeast Asia".

49: "...from a corpus of 15,426 publications..." this isn't necessary for the abstract and should be removed.

50: "1.5 times"? But your results say 65% increase.

85: "loss of biodiversity" – I'm not sure why you haven't cited Gibson et al. Nature 2011 DOI: 10.1038/nature10425, the primary reference for this statement. Add it.

94: "biodiversity loss" – once again, you are omitting the key reference to this statement, Gibson et al. Nature 2011.

96-99: This has been examined. See Guo et al. EcoHealth DOI: 10.1007/s10393-018-1336-3

107: "between land-use and land-use change or biodiversity loss" I think you mean "disease risks" somewhere. Again, you have omitted a key reference and should cite Guo et al. EcoHealth (see above).

133: "two reviewers" who? (the same as in line 160?) Or would you rather say "we"?

295, 387, 422, 435, 440, 627: "palm oil" is the product, "oil palm" is the tree crop. So these cases should all be changed to "oil palm" / "oil palm plantation"

414-415: cite Guo et al. EcoHealth

492: change period to comma

Reviewer #2:

Remarks to the Author:

The authors have used a review methodology quite popular with the proliferation of publication in science, to explore hypotheses around a plausible association between human disease incidence and land use, in particular agriculture. The authors also recognise the potential danger of this method which can have extreme bias and confounding as it is attempting to generalise across a highly complex socioecology. The resulting article methodology is also almost incomprehensible, with its attempt to justify and resolve the pitfalls with such an approach. The analysis of a very few articles ~ 34 from a large number (over 15000) which the search strings brought up, is evidence of the challenge to provide believable results using this approach. The authors should be commended with the rigour with which they have approached the subject and method but there is a danger of missing the wood for the trees. On the one hand, this approach is something to be discouraged at the nexus of science and policy, as the reader in many instances is not reading the complexity, just the conclusions and the interpretation can in itself be biased leading to quite unjustified influence on the perception of the general public, policy makers and others of this sort of "hands off" science. Another example of this approach was the Jones et al 2008 paper on emerging diseases, which led to a proliferation of the idea that pretty much all human disease comes from animals and much of this from wildlife. Clearly not the objective of the review which in reality was inherently very biased as the research was based on one journal only and only ~ 40 years of publications. Similar approaches to correcting bias etc were made in that article to this one. Although the conclusions were justified based on the criteria set and even from their small sample, but as generalisable fact, the conclusions are hardly justifiable in my opinion, based on the method. Ironically, in the Jones paper the conclusion might well be a truth as reflection on the question suggests that it is a fair assumption that emergent human disease must come from somewhere (otherwise it would be there already) and, the only place it could come from is biodiversity, unless you believe theories of new pathogens arriving by meteorite which is a strongly held view by some. So in this case the authors successfully stated the obvious but it was somehow more believable wrapped up in the complex methodologies. The negative side to this result is that perceptions are that wildlife and nature are a threat, somehow reinforcing prejudice against nature that has existed for millennia, over the time of human development. Policy that might be destructive does arise from this, which is totally unjustified and counterproductive. This paper is less crude in its approach and is more comprehensive from literature and focused on a specific land use change, which has been ongoing for only 20,000s years, approximately, with the advent of agriculture in human society. The rate of change has accelerated in the past century associated with the human population explosion so its impacts are now increasingly apparent and externalities such as emergent disease driven by the process can be explored, risk pathways and mechanisms understood. So a key question that I have is, how will this result be interpreted, will it be seen as a threat to agriculturalists and the development paradigm? Rather like papers on livestock and climate change, whole of society influences can arise from the science which is often taken out of context and simplified to justify points of view which can lead to real dangers in crude policy initiative and regulation. So although I am impressed by the level of detail in this paper and the efforts made I am more concerned about how the outcomes are interpreted and the inferences made in the discussion and conclusion. I accept the authors try to not overstate the evidence but I am afraid readers will not read that aspect in the text just their own interpretation of the final wording and even the title: "Agricultural land-uses consistently exacerbate infectious disease risks in South East Asia." When in fact I suggest what you are saying is that a list of certain diseases, that had sufficient data method etc published in English to interpret, appear to be associated with a person's association with that land use and location. A sort of occupational or geographic risk which is of course not new. For example, we have long known that

slaughter house workers have a higher incidence of certain diseases, so the principle idea is not wrong but we cannot generalise in my opinion to the full title of the paper from these data. This will be misinterpreted I suspect to a point that is not supported by the paper. So my first request is to change the title to be more accurate to the facts. An example of an acceptable change could be "...exacerbate certain infectious disease risks...." What makes me a bit suspicious also, that the method, is to some extent inherently flawed (and therefore incapable of providing absolute truth), is that no virus infections came up on the list, when we know that most emergent diseases in the last century are viruses. You mention in the introduction Nipah virus, with a close association with agriculture and environmental change yet it does not emerge from literature in the way it might. The same might be said for Japanese Encephalitis virus see...

<https://www.sciencemag.org/news/2016/02/japanese-encephalitis-could-have-new-transmission-route-pigs>. This may well not be the fault of the principle method but more to do with the quality of data available, which the authors point out but this does not justify the overall approach, the danger is in fact rubbish in, rubbish out. However, more specifically on this point I would ask that you review possible reasons why virus studies are not coming out of the woodwork, in support of the hypotheses. Are the studies so badly done? Although you do defend the work robustly, I would like to see more of the concern that you show in sub-sections on the reliability of the results, and interpreted more generally in the discussion and conclusion. I would also have liked to see some triangulation with other methods such as participatory epidemiology, which is very poorly represented in literature, and a further effort to access non-English text from the medical community in the region, to test some of these conclusions. A mixed methods approach in fact I believe would be more robust. Some few specific comments are made on the paper in sticky notes. I would also strongly recommend that you consider shortening the article, with the bulk of the statistical material, figures and table in a supplementary file but with more careful translation of the outcomes of those efforts in the discussion and conclusions as stated above.

Review 1

General response: We would like to thank the reviewer for their constructive and well supported feedback. We have addressed all the comments below, added key citations that were highlighted and made all other requested changes to the manuscript.

Comment 1 - I would've liked to see more about the relationship between the different variables (e.g., crop type vs. disease type), see Gibson et al. Nature Figure 3 (which should be cited, see below) for a good example. Can you add something examining the relationship between different variables?

Response 1 – This is an excellent suggestion. We have now included additional subgroup analyses that assess the association between exposure subgroups and additional disease classes - both aetiological agent (parasitic, viral, bacterial) and transmission mode (vector-borne, zoonotic) are now considered, as well as additional resolution on livestock exposure categories (all livestock, porcine/bovine/poultry). This analysis is now presented using 3 figures (Figure 4 for agricultural exposure subgroups, Figure 5 for livestock-based subgroups and Figure 6 for disease-based subgroups). Note that in order to preserve sample sizes, transmission mode subgroups are not mutually exclusive (e.g., a disease can be both vector-borne and zoonotic, such as zoonotic malaria) but nevertheless the analysis usefully illustrates the variation (or lack thereof) in the responses between these groups. This revised analysis provides additional information to the original analyses.

While we still see clear associations for forest monoculture-based land use (oil palm plantation and rubber) and marginal associations for rice paddy farming, clear positive associations are now evident for livestock farming albeit with variation by aetiological agent. This analysis also clarifies responses for viral diseases (H5N1, Nipah virus and Hepatitis E virus) raised in Review 2 (see below). The additional results have been fully described in the results sections from lines 207-251. Additionally, we updated our discussion based on our new findings, which can be found from lines 272 – 376 and lines 319 – 341. In lines 410 – 413, we also provide an explicit statement for readers to note small sample sizes within some of the subgroups and caution against generalisations.

Comment 2 - What is going on with the Methods section? You have 3 parts – the first “Search Strategy and selection process” using normal sentences, the second “Eligibility” as a list, the third “Inclusion Criteria” as bullet points – all with the same information. Only one is needed.

Response 2 – This is useful feedback and we have now revised the methods section to reflect. However, as per PRISMA reporting standards, we are obliged to be transparent with regards to our methods for systematic reviewing and to include the information required to conform to the PRISMA guidelines. As such, we have 1) kept the “Search Strategy and selection process” and the PECOS framework list in “Eligibility” in the main text, and 2) moved the example search strategy and bulleted inclusion and exclusion criteria to the Supplementary Information for brevity and to avoid duplication in the main text.

Comment 3 - 2, 49, 118, 434, 444, 532: “South-East Asia” or “South East Asia” or “south-east Asia”? Be consistent. The most commonly used is something else: “Southeast Asia”.

Response 3 – This has been amended to Southeast Asia in the manuscript and abbreviated to SE Asia where appropriate.

Comment 4 - 49: “...from a corpus of 15,426 publications...” this isn't necessary for the abstract and should be removed.

Response 4 – This has been amended in the manuscript.

Comment 5 - 50: “1.5 times”? But your results say 65% increase.

Response 5 – This has been amended in the manuscript.

Comment 6 - 85: “loss of biodiversity” – I’m not sure why you haven’t cited Gibson et al. Nature 2011 DOI: 10.1038/nature10425, the primary reference for this statement. Add it.

Response 6 – This key reference is now cited in several places in the manuscript.

Comment 7 - 94: “biodiversity loss” – once again, you are omitting the key reference to this statement, Gibson et al. Nature 2011.

Response 7 – As above.

Comment 8 - 96-99: This has been examined. See Guo et al. EcoHealth DOI: 10.1007/s10393-018-1336-3

Response 8 – We thank the reviewer for bringing this important study to our attention, and we have now cited and elaborated on this study accordingly (lines 122-125; see also comment 9/response 9).

Comment 9 - 107: “between land-use and land-use change or biodiversity loss” I think you mean “disease risks” somewhere. Again, you have omitted a key reference and should cite Guo et al. EcoHealth (see above).

Response 9 – As above. We have also revised the sentence for clarity (lines 119-125).

Comment 10 - 133: “two reviewers” who? (the same as in line 160?) Or would you rather say “we”?

Response 10 – This has been amended in the manuscript. We now use “we” (line 505).

Comment 11 - 295, 387, 422, 435, 440, 627: “palm oil” is the product, “oil palm” is the tree crop. So these cases should all be changed to “oil palm” / “oil palm plantation”

Response 11 – Good clarification. We have amended the manuscript accordingly.

Comment 12 - 414-415: cite Guo et al. EcoHealth

Response 12 – As above, this study has been cited accordingly.

Comment 13 - 492: change period to comma

Response 13 – This has been amended in the manuscript.

Review 2

General response: We would like to thank the reviewer for their very valuable and thoughtful comments prior to making their more specific suggestions outlined below. We have reflected on these comments and have taken all opportunities to clarify the manuscript and to stick as close to the data as possible as requested. With reference to the comparison made with other studies (e.g.

Jones et al., 2008) and the issues raised with respect to the potential for misinterpretation due to data availability, data retrieval, bias, heterogeneity and so on, we would also emphasise here that a key difference between ours and previous studies on related topics is that we have sought to objectively apply the gold standard methodology in the medical sciences for quantitative systematic review and meta-analysis to detect potential issues and limit the extent to which they could result in misinterpretation. More specific details for each point raised are provided below.

Comment 1 - change the title to be more accurate to the facts. An example of an acceptable change could be "...exacerbate certain infectious disease

Response 1 – We thank the reviewer for this suggestion and have carefully considered the concern expressed regarding the accuracy of the title. We in no way intended to overstate or inaccurately reflect our results, with the original title simply reflecting the overall (net) effect calculated from the meta-analysis across all studies included. After careful consideration, we feel that this remains the most appropriate 'high level' finding to report in the title and that modifying the title as suggested would be somewhat inconsistent with the overall objective of evaluating an overall effect size via rigorous and consistent systematic review and meta-analysis methodology. With that said, we completely agree with the reviewer that any risk that our contribution could be misinterpreted as stating that agricultural exposure exacerbates ALL infectious disease risks must be eliminated. We do not however believe we have implied this anywhere in the manuscript and in contrast make this completely clear as early as the Abstract, in which we provide a clear portrayal of our methodology and results, including statements on the main diseases and subgroups that have a significant positive association and those that do not. We further reflect on this issue in more detail in each subsequent section. As such, we propose to modify the title only to conform to the Nature Communications title format requirements of no punctuation (we have removed the colon and subsequent text "": a systematic review and meta-analysis").

Comment 2 - I would ask that you review possible reasons why virus studies are not coming out of the woodwork, in support of the hypotheses. Are the studies so badly done?

Response 2 – This was an insightful comment that prompted us to reconsider our results and ultimately include an additional analysis to address. In our initial analysis, we did not include separate subgroup analyses to distinguish between different disease classes (aetiological agent, transmission mode) and therefore any association with viral diseases was not made explicit. However, as can be seen in Fig 1, several studies that addressed viral diseases were retained for analysis after meeting the inclusion / exclusion criteria (e.g., H5N1, Nipah virus, Hepatitis E). In this revision, we have updated our analysis to include subgroup analyses for different disease groups (aetiological agent: parasitic, viral, bacterial; transmission mode: zoonotic, vector-borne), also in response to a comment in Review 1. In this new analysis, we see a strong association for livestock farming and most disease classes, and we found a marginal positive association between livestock farming and viral diseases consistent with the overall trend. However, sample size is eroded at this level of subgrouping, so confidence intervals are relatively wide and in this case result in non-significance for this subgrouping. We include description of the new analysis and results from lines 207-251 and discussion from lines 272 – 376 and lines 319 – 341. In lines 410 – 413, we also provide an explicit statement for readers to take note of small sizes in some subgroups.

In relation to the viral studies retained in our analyses, only 3 studies from the original corpus were considered eligible based on our strict inclusion and exclusion criteria. All 3 studies are classified as 'livestock farming'. Studies on Japanese encephalitis were retrieved by the search criteria but ultimately excluded as all were epidemiological studies that assessed residential or occupational exposure and JE prevalence or incidence in children (for our study, studies that explicitly assess a child population aged 18 and under were excluded to ensure a homogenous dataset for generalisability). To clarify, our inclusion criteria states that studies that recruited participants of all

ages (including children) were included, however, studies that focused exclusively on the child population were excluded such as Liu et al, 2010 (Risk factors for Japanese encephalitis: a case-control study).

Similarly, studies on Nipah Virus were retrieved from the search criteria but many were ultimately excluded due to their focus on occupational exposure in slaughterhouses, which we considered 'agriculture manufacture' (occurs in a factory) as opposed to agricultural production which involves exposure to agricultural land use or land use change. In addition, many Nipah studies make use of the same dataset from hospital based cross-sectional data (e.g. Amal et al (2000); Parashar et al (2000)) and therefore we only included one study in our analysis to avoid duplication and the introduction of bias, as described in the Methods. No other studies on viral diseases were retained for final analysis.

Comment 3: Although you do defend, the work robustly, I would like to see more of the concern that you show in sub-sections on the reliability of the results and interpreted more generally in the discussion and conclusion.

Response 3: The reviewer correctly raises an important issue on the reliability and generalisability of our results. Our methodology is the gold standard method for systematic review and meta-analysis which by itself, partly reduces bias and allows for transparency, thereby leading to more accurate and reliable conclusions than might otherwise be the case from a bespoke or non-systematic review methodology. We also include a range of additional tests for confounding, bias and heterogeneity to further improve our ability to interpret the results. In addition, the systematic review methodology is also intended to make the information easier for the end user to read and understand. Using this methodology, we have consistently highlighted the bias, confounding and effect modification that is possible with such distal risk factors throughout the manuscript. We have also signposted the reader accordingly to additional research that further supports our findings and provide recommendations on future research to improve the evidence base for such nexus issues.

However, to provide further clarification to readers, we now incorporate an explicit statement to note small sample sizes within some of the subgroups and caution against generalisations not supported by the data (lines 410 – 413). In addition, we also provide recommendations to readers to generate better quality robust evidence in the form of empirical data from appropriately powered epidemiological studies that assess occupational or residential exposure to agriculture/livestock with the risk of infectious disease (see Review 2, comment 2, response 2). We also state “we identify a general paucity of the highest quality studies on the human health implications of land-use decision making and policy, and its impacts on infectious diseases. Further studies that capture bias, confounding and effect modification would be particularly valuable.” (see lines 418-422). In addition, our manuscript has already previously highlighted the potential limitations regarding confounding, effect modification, lack of robust high-quality evidence and spatio-temporal heterogeneity which can all be found in the discussion.

Comment 4 - So a key question that I have is, how will this result be interpreted, will it be a threat to agriculturalists and the development paradigm? Rather like papers on livestock and climate change, whole of society influences can arise from the science which is often taken out of context and simplified to justify points of view which can lead to real dangers in crude policy initiative and regulation. So, although I am impressed by the level of detail in this paper and the efforts made, I am more concerned about how the outcomes are interpreted and the inferences made in the discussion and conclusion.

Response 4 – The reviewer raises an important point about how our findings will be, should be or could be interpreted by members of the agricultural and development sectors (and indeed any other readers). In principal, we view our contribution as providing much needed scientific data on

questions relevant to both sectors and have gone to great lengths to provide objective, transparent, gold-standard methodology and analysis to help interpret these data. We are not prescriptive to any stakeholder in particular, but rather envisage that various stakeholder groups will find these data and analyses of use and interest for their own respective needs and purposes. While our intention is not to be inflammatory and we have gone to great lengths to help facilitate robust interpretation, we recognise that interests may not always align and there is potential, as for any study, for our results to be taken out of context. We have thus redoubled our efforts to provide clear and transparent interpretations throughout the manuscript. For example, we now include explicit statements in lines 410 – 413 regarding the issue of generalisability and interpretation of our results. In addition, in our final paragraph, we provide our evidence-based synthesis that there is an opportunity here for policy and decision makers to consider our results on infectious diseases alongside other documented benefits and impacts of agriculture and agricultural development in exposed populations and affected environments. On the strength of our results, we go on to argue that considered investment and collaboration among public health and sustainable development stakeholders could play a pivotal role in resolving and mitigating such negative externalities and achieving other related goals (e.g., avoiding or reversing biodiversity loss, minimising GHG emissions, ecosystem service protection).

Comment 5 - I would also have liked to see some triangulation with other methods such as participatory epidemiology, which is very poorly represented in literature, and a further effort to access non-English text from the medical community in the region, to test some of these conclusions. A mixed methods approach in fact I believe would be more robust.

Response 5 – We fully agree with this suggestion for the use and importance of other available methods such as participatory epidemiology in the surveillance, control and prevention of infectious disease in humans and animals. Although it is beyond the scope of the current study to do this given that it is a broad scale systematic review of the literature covering 12 exposures and 26 infectious diseases or 6 disease complexes, we do now provide recommendations in the discussion for the use of participatory epidemiology as a tool for bottom up agro-system analytical research (see lines 434 to 444). We also provide examples of H5N1 in Egypt and rinderpest control, which used participatory methods to elucidate key insights and signpost readers to the respective citations.

With respect to non-English texts, as stated below in comment 6, we would like to make clear that we did not include any language restrictions in the search strategy to ensure it as sensitive and specific as possible to capture all relevant articles that fitted the research question.

However, once we had screened all articles, we found zero non-English language articles that could have been included and that met the eligibility criteria. Surprisingly, we similarly found zero articles that were not in English through snowballing of references, suggesting that the extent and effects of language bias may have diminished because of the shift towards publication of studies in English. This concurs with the guidance provided in the Cochrane Handbook of Systematic Review and Meta-Analysis (which we have cited accordingly). We have stated the above in lines 523– 526. We have also specified that no language restrictions were placed in the search strategy on line 502.

Comment 6- I would also strongly recommend that you consider shortening the article, with the bulk of the statistical material, figures and table in a supplementary file but with more careful translation of the outcomes of those efforts in the discussion and conclusions as stated above.

Response 6 – We welcome this suggestion of brevity and have adapted our figures accordingly. As suggested, we have moved all statistical information into the appendices. We have also amalgamated the exposure and disease-based subgroup and the livestock subgroup analysis to form 3 informative figures. Furthermore, we have shifted a large section of methods to the supplementary information (see Review 1, Comment 2 above).

Comment 7- Line 110 - Another review paper might of interest here...Kock R. (2015) Vertebrate reservoirs and secondary epidemiological cycles of vector-borne diseases Rev. Sci. Tech. Off. Int. Epiz., 2015, 34 (1): 151-163

Response 7 – This is indeed a valuable reference, which we now cite. We have also added specific case studies from this paper regarding Japanese encephalitis and Culex mosquitoes on line 111-114. Specifically, we state “For example, irrigation-based agriculture and rural development have expanded breeding habitats of Culex vectors and has led to Japanese encephalitis virus establishing a secondary cycle in domestic pig populations where it amplifies and spills over into human populations.”

Comment 8- Line 120 - This is a pity especially given the value of looking at less well agriculturally developed landscapes as occur in Africa to test earlier stages in this trend. I think this is a substantial weakness of the review and it is not clear why exactly this was considered infeasible. I suggest more explanation for this would be valuable.

Response 8 – We agree that it would be of interest to look at other regions to formulate a true ‘global’ analysis. However, based on preliminary work, a global study was deemed infeasible given the resources and time that would have been required to double review more than 50,000 studies within the time frame we were able to allocate to this study. We subsequently selected SE Asia as a more manageable model system that remained well suited to still adequately address our hypotheses. We have now clarified our justification on line 172-182. We have also added a citation that supports SE Asia as an appropriate model system for investigating issues at the nexus of biodiversity conservation, infectious disease and sustainable development (Coleman et al, 2019).

Comment 9 - Line 174 - this needs justification as presumably it excludes a considerable amount of data?

Response 9 – As above (comment 5)

Comment 10 - Line 234 and 233 - large text size

Response 10 – This has been amended in the manuscript.

Comment 11 - Line 340 - But I understood only articles in English were accessed?

Response 11 – As above (response 6) - we included no restrictions on language within the search strategy and we found zero non-English language articles that could have been included and that met the eligibility criteria.

Comment 12 - Line 496 - Hong Kong

Response 12 – This has been amended within the manuscript.

Comment 13 - Line 609 - I believe this should be balanced with a statement on modifications to demand that are implicit in many policy changes in the pipeline e.g. on food waste currently proposed by international agencies, reduction in KCAL intake recommended by many health authorities etc. soil capping regulations in the EC soil directive, reductions in animal products consumed that is advised to mitigate climate change effects of livestock etc. There are many reasons to stop increasing agricultural production and land conversion, but we remain stuck in the food security paradigm which was a post war product! The assumptions implicit in this statement

perhaps should be critically examined in light of current trends. This publication's findings further support and justify a trend away from change in land use to agriculture and rather puts emphasis on the urgent need for recovery of ecosystems and attention to environmental resilience to reduce negative externalities in this case disease risk.

Response 13 – We thank the reviewer for steering us in disseminating further tangible conclusions from our research and are pleased with the support and focus on the issue of co-benefits. We have now referred to this in the discussion (lines 494-514). Specifically, we have incorporated a final paragraph that recognises the negative externalities of agricultural land use, current policies that are already being conducted to aid in mitigating these externalities and the potential opportunity for ecosystem recovery and restoration to generate co-benefits, including the consideration of health impacts.

Reviewers' Comments:

Reviewer #1:

Remarks to the Author:

The authors have made a strong revision of the manuscript and I now find it suitable for publication in Nature Communications. Nicely done!

Reviewer #2:

Remarks to the Author:

The paper is a systematic review and meta-analysis of available literature with rigorous inclusion and exclusion criteria according to Prisma protocol and reporting standards, examining the influence of agricultural land use on human infectious diseases occurrence in South East Asia. The findings suggest significantly higher risk from disease amongst humans living in agricultural landscapes for certain diseases, and especially in association with certain crops and a strong association with livestock. The authors have used every tool available to ensure transparency and rigorous method to avoid bias and misinterpretation of the results, which is vital given known ecologically complex relationships between host, pathogen, vectors and environment. The concerns in first review have been adequately addressed and the paper now conforms to the standards expected of Nature Communications and current scientific practice in this field of enquiry and the conclusions are sound.

REVIEWERS' COMMENTS:

Reviewer #1 (Remarks to the Author):

The authors have made a strong revision of the manuscript and I now find it suitable for publication in Nature Communications. Nicely done!

Reviewer #2 (Remarks to the Author):

The paper is a systematic review and meta-analysis of available literature with rigorous inclusion and exclusion criteria according to Prisma protocol and reporting standards, examining the influence of agricultural land use on human infectious diseases occurrence in South East Asia. The findings suggest significantly higher risk from disease amongst humans living in agricultural landscapes for certain diseases, and especially in association with certain crops and a strong association with livestock. The authors have used every tool available to ensure transparency and rigorous method to avoid bias and misinterpretation of the results, which is vital given known ecologically complex relationships between host, pathogen, vectors and environment. The concerns in first review have been adequately addressed and the paper now conforms to the standards expected of Nature Communications and current scientific practice in this field of enquiry and the conclusions are sound.